# Can We Gain More from Orthogonality Regularizations in Training Deep CNNs?

**Nitin Bansal**     **Xiaohan Chen**     **Zhangyang Wang**

Department of Computer Science and Engineering
Texas A&M University,
College Station, TX 77843, USA
{bansa01, chernxh, atlaswang}@tamu.edu

## Abstract

This paper seeks to answer the question: *as the (near-) orthogonality of weights is found to be a favorable property for training deep convolutional neural networks, how can we enforce it in more effective and easy-to-use ways?* We develop novel orthogonality regularizations on training deep CNNs, utilizing various advanced analytical tools such as mutual coherence and restricted isometry property. These plug-and-play regularizations can be conveniently incorporated into training almost any CNN without extra hassle. We then benchmark their effects on state-of-the-art models: ResNet, WideResNet, and ResNeXt, on several most popular computer vision datasets: CIFAR-10, CIFAR-100, SVHN and ImageNet. We observe consistent performance gains after applying those proposed regularizations, in terms of both the final accuracies achieved, and faster and more stable convergences. We have made our codes and pre-trained models publicly available.[1].

## 1   Introduction

Despite the tremendous success of deep convolutional neural networks (CNNs) [1], their training remains to be notoriously difficult both theoretically and practically, especially for state-of-the-art ultra-deep CNNs. Potential reasons accounting for such difficulty lie in multiple folds, ranging from vanishing/exploding gradients [2], to feature statistic shifts [3], to the proliferation of saddle points [4], and so on. To address these issues, various solutions have been proposed to alleviate those issues, examples of which include parameter initialization [5], residual connections [6], normalization of internal activations [3], and second-order optimization algorithms [4].

This paper focuses on one type of structural regularizations: *orthogonality*, to be imposed on linear transformations between hidden layers of CNNs. The orthogonality implies energy preservation, which is extensively explored for filter banks in signal processing and guarantees that energy of activations will not be amplified [7]. Therefore, it can stabilize the distribution of activations over layers within CNNs [8, 9] and make optimization more efficient. [5] advocates orthogonal initialization of weight matrices, and theoretically analyzes its effects on learning efficiency using deep linear networks. Practical results on image classification using orthogonal initialization are also presented in [10]. More recently, a few works [11–15] look at (various forms of) enforcing orthogonality regularizations or constraints throughout training, as part of their specialized models for applications such as classification [14] or person re-identification [16]. They observed encouraging result improvements. However, a dedicated and thorough examination on the effects of orthogonality for training state-of-the-art general CNNs has been absent so far.

Even more importantly, how to evaluate and enforce orthogonality for non-square weight matrices does not have a sole optimal answer. As we will explain later, existing works employ the most obvious but not necessarily appropriate option. We will introduce a series of more sophisticated regularizers that lead to larger performance gains.

This paper investigates and pushes forward various ways to enforce orthogonality regularizations on training deep CNNs. Specifically, we introduce three novel regularization forms for orthogonality, ranging from the double-sided variant of standard Frobenius norm-based regularizer, to utilizing Mutual Coherence (MC) and Restricted Isometry Property (RIP) tools [17–19]. Those orthogonality regularizations have a plug-and-play nature, i.e., they can be incorporated with training almost any CNN without hassle. We extensively evaluate the proposed orthogonality regularizations on three state-of-the-art CNNs: ResNet [6], ResNeXt [20], and WideResNet [21]. In all experiments, we observe the consistent and remarkable accuracy boosts (e.g., **2.31%** in CIFAR-100 top-1 accuracy for WideResNet), as well as faster and more stable convergences, *without any other change made to the original models*. It implies that many deep CNNs may have not been unleashed with their full powers yet, where orthogonality regularizations can help. Our experiments further reveal that larger performance gains can be attained by designing stronger forms of orthogonality regularizations. We find the RIP-based regularizer, which has better analytical grounds to characterize near-orthogonal systems [22], to consistently outperform existing Frobenius norm-based regularizers and others.

## 2   Related Work

To remedy unstable gradient and co-variate shift problems, [2, 23] advocated near constant variances of each layer's output for initialization. [3] presented a major breakthrough in stabilizing training, via ensuring each layer's output to be identical distributions which reduce the internal covariate shift. [24] further decoupled the norm of the weight vector from its phase(direction) while introducing independences between minibatch examples, resulting in a better optimization problem. Orthogonal weights have been widely explored in Recurrent Neural Networks (RNNs) [25–30] to help avoid gradient vanishing/explosion. [25] proposed a soft constraint technique to combat vanishing gradient, by forcing the Jacobian matrices to preserve energy measured by Frobenius norm. The more recent study [29] investigated the effect of soft versus hard orthogonal constraints on the performance of RNNs, the former by specifying an allowable range for the maximum singular value of the transition matrix and thus allowing for its small intervals around one.

In CNNs, orthogonal weights are also recognized to stabilize the layer-wise distribution of activations [8] and make optimization more efficient. [5, 10] presented the idea of orthogonal weight initialization in CNNs, which is driven by the norm-preserving property of orthogonal matrix: a similar outcome which BN tried to achieve. [5] analyzed the non-linear dynamics of CNN training. Under simplified assumptions, they concluded that random orthogonal initialization of weights will give rise to the same convergence rate as unsupervised pre-training, and will be superior than random Gaussian initialization. However, a good initial condition such as orthogonality does not necessarily sustain throughout training. In fact, the weight orthogonality and isometry will break down easily when training starts, if not properly regularized [5]. Several recent works [12, 13, 15] considered Stiefel manifold-based hard constraints of weights. [12] proposed a Stiefel layer to guarantee fully connected layers to be orthogonal by using Reimannian gradients, without considering similar handling for convolutional layers; their performance reported on VGG networks [31] were less than promising. [13] extended Riemannian optimization to convolutional layers and require filters within the same channel to be orthogonal. To overcome the challenge that CNN weights are usually rectangular rather than square matrices, [15] generalized Stiefel manifold property and formulated an Optimization over Multiple Dependent Stiefel Manifolds (OMDSM) problem. Different from [13], it ensured filters across channels to be orthogonal. A related work [11] adopted a Singular Value Bounding (SVB) method, via explicitly thresholding the singular values of weight matrices between a pre-specified narrow band around the value of one.

The above methods [11–13, 15] all fall in the category of enforcing "hard orthogonality constraints" into optimization ([11] could be viewed as a relaxed constraint), and have to repeat singular value decomposition (SVD) during training. The cost of SVD on high-dimensional matrices is expensive even in GPUs, *which is one reason why we choose not to go for the "hard constraint" direction in this paper*. Moreover, since CNN weight matrices cannot exactly lie on a Stiefel manifold as they are either very "thin" or "fat" (e.g., $W^T W = I$ may never happen for an overcomplete "fat" $W$ due to

rank deficiency of its gram matrix), special treatments are needed to maintain the hard constraint. For example, [15] proposed group based orthogonalization to first divide an over-complete weight matrix into "thin" column-wise groups, and then applying Stiefel manifold constraints group-wise. The strategy was also motivated by reducing the computational burden of computing large-scale SVDs. Lately, [32, 33] interpreted CNNs as Template Matching Machines, and proposed a penalty term to force the templates to be orthogonal with each other, leading to significantly improved classification performance and reduced overfitting with no change to the deep architecture.

A recent work [14] explored orthogonal regularization, by enforcing the Gram matrix of each weight matrix to be close to identity under Frobenius norm. It constrains orthogonality among filters in one layer, leading to smaller correlations among learned features and implicitly reducing the filter redundancy. Such a soft orthonormal regularizer is differentiable and requires no SVD, thus being computationally cheaper than its "hard constraint" siblings. However, we will see later that Frobenius norm-based orthogonality regularization is only a rough approximation, and is inaccurate for "fat" matrices as well. The authors relied on a backward error modulation step, as well as similar group-wise orthogonalization as in [15]. We also notice that [14] displayed the strong advantage of enforcing orthogonality in training the authors' self-designed plain deep CNNs (i.e. without residual connections). However, they found fewer performance impacts when applying the same to training prevalent network architectures such as ResNet [6]. In comparison, our orthogonality regularizations can be added to CNNs as "plug-and-play" components, without any other modification needed. We observe evident improvements brought by them on most popular ResNet architectures.

Finally, we briefly outline a few works related to orthogonality in more general senses. One may notice that enforcing matrix to be (near-)orthogonal during training will lead to its spectral norm being always equal (or close) to one, which links between regularizing orthogonality and spectrum. In [34], the authors showed that the spectrum of Extended Data Jacobian Matrix (EDJM) affected the network performance, and proposed a spectral soft regularizer that encourages major singular values of EDJM to be closer to the largest one. [35] claimed that the maximum eigenvalue of the Hessian predicted the generalizability of CNNs. Motivated by that, [36] penalized the spectral norm of weight matrices in CNNs. A similar idea was later extended in [37] for training generative adversarial networks, by proposing a spectral normalization technique to normalize the spectral norm/Lipschitz norm of the weight matrix to be one.

## 3 Deriving New Orthogonality Regularizations

In this section, we will derive and discuss several orthogonality regularizers. Note that those regularizers are applicable to both fully-connected and convolutional layers. The default mathematical expressions of regularizers will be assumed on a fully-connected layer $W \in {}^{m \times n}$ ($m$ could be either larger or smaller than $n$). For a convolutional layer $C \in {}^{S \times H \times C \times M}$, where $S, H, C, M$ are filter width, filter height, input channel number and output channel number, respectively, we will first reshape $C$ into a matrix form $W' \in m' \times n'$, where $m' = S \times H \times C$ and $n' = M$. The setting for regularizing convolutional layers follows [14, 15] to enforces orthogonality across filter, encouraging filter diversity. All our regularizations are directly amendable to almost any CNN: there is no change needed on the network architecture, nor any other training protocol (unless otherwise specified).

### 3.1 Baseline: Soft Orthogonality Regularization

Previous works [14, 32, 33] proposed to require the Gram matrix of the weight matrix to be close to identity, which we term as Soft Orthogonality (SO) regularization:

$$\text{(SO)} \qquad \lambda ||W^T W - I||_F^2, \tag{1}$$

where $\lambda$ is the regularization coefficient (the same hereinafter). It is a straightforward relaxation from the "hard orthogonality" assumption [12, 13, 15, 38] under the standard Frobenius norm, and can be viewed as a different weight decay term limiting the set of parameters close to a Stiefel manifold rather than inside a hypersphere. The gradient is given in an explicit form: $4\lambda W(W^T W - I)$, and can be directly appended to the original gradient w.r.t. the current weight $W$.

However, SO (1) is flawed for an obvious reason: the columns of $W$ could possibly be mutually orthogonal, if and only if $W$ is undercomplete ($m \geq n$). For overcomplete $W$ ($m < n$), its gram matrix $W^T W \in \mathbb{R}^{n \times n}$ cannot be even close to identity, because its rank is at most $m$, making

$||W^T W - I||_F^2$ a biased minimization objective. In practice, both cases can be found for layer-wise weight dimensions. The authors of [15, 14] advocated to further divide overcomplete $W$ into undercomplete column groups to resolve the rank deficiency trap. In this paper, we choose to simply use the original SO version (1) as a fair comparison baseline.

The authors of [14] argued against the hybrid utilization of the original $\ell_2$ weight decay and the SO regularization. They suggested to stick to one type of regularization all along training. Our experiments also find that applying both together throughout training will hurt the final accuracy. Instead of simply discarding $\ell_2$ weight decay, we discover a *scheme change* approach which is validated to be most beneficial to performance, details on this can be found in Section 4.1.

## 3.2 Double Soft Orthogonality Regularization

The double soft orthogonality regularization extends SO in the following form:

$$\text{(DSO)} \qquad \lambda(||W^T W - I||_F^2 + ||W W^T - I||_F^2). \qquad (2)$$

Note that an orthogonal $W$ will satisfy $W^T W = W W^T = I$; an overcomplete $W$ can be regularized to have small $||W W^T - I||_F^2$ but will likely have large residual $||W^T W - I||_F^2$, and vice versa for an under-complete $W$. DSO is thus designed to cover both over-complete and under-complete $W$ cases; for either case, at least one term in (2) can be well suppressed, requiring either rows or columns of $W$ to stay orthogonal. It is a straightforward extension from SO.

Another similar alternative to DSO is "selective" soft orthogonality regularization, defined as: $\lambda ||W^T W - I||_F^2$, if $m > n$; $\lambda ||W W^T - I||_F^2$ if $m \leq n$. Our experiments find that DSO always outperforms the selective regularization, therefore we only report DSO results.

## 3.3 Mutual Coherence Regularization

The mutual coherence [18] of $W$ is defined as:

$$\mu_W = \max_{i \neq j} \frac{|\langle w_i, w_j \rangle|}{||w_i|| \cdot ||w_j||}, \qquad (3)$$

where $w_i$ denotes the $i$-th column of $W$, $i = 1, 2, ..., n$. The mutual coherence (3) takes values between [0,1], and measures the highest correlation between any two columns of $W$. In order for $W$ to have orthogonal or near-orthogonal columns, $\mu_W$ should be as low as possible (zero if $m \geq n$).

We wish to suppress $\mu_W$ as an alternative way to enforce orthogonality. Assume $W$ has been first normalized to have unit-norm columns, $\langle w_i, w_j \rangle$ is essentially the $(i, j)$-the element of the Gram matrix $W^T W$, and $i \neq j$ requires us to consider off-diagonal elements only. Therefore, we propose the following mutual coherence (MC) regularization term inspired by (3:

$$\text{(MC)} \qquad \lambda ||W^T W - I||_\infty. \qquad (4)$$

Although we do not explicitly normalize the column norm of $W$ to be one, we find experimentally that minimizing (4) often tends to implicitly encourage close-to-unit-column-norm $W$ too, making the objective of (4) a viable approximation of mutual coherence (3)[2].

The gradient of $||W^T W - I||_\infty$ could be explicitly solved by applying a smoothing technique to the nonsmooth $\ell_\infty$ norm, e.g., [39]. However, it will invoke an iterative routine each time to compute $\ell_1$-ball proximal projection, which is less efficient in our scenario where massive gradient computations are needed. In view of that, we turn to using auto-differentiation to approximately compute the gradient of (4) w.r.t. $W$.

## 3.4 Spectral Restricted Isometry Property Regularization

Recall that the RIP condition [17] of $W$ assumes:

**Assumption 1** *For all vectors $z \in \mathbb{R}^n$ that is $k$-sparse, there exists a small $\delta_W \in (0, 1)$ s.t. $(1 - \delta_W) \leq \frac{||Wz||^2}{||z||^2} \leq (1 + \delta_W)$.*

The above RIP condition essentially requires that every set of columns in $W$, with cardinality no larger than $k$, shall behave like an orthogonal system. If taking an extreme case with $k = n$, RIP then turns into another criterion that enforces the entire $W$ to be close to orthogonal. Note that both mutual incoherence and RIP are well defined for both under-complete and over-complete matrices.

We rewrite the special RIP condition with $k = n$ in the form below:

$$\left| \frac{||Wz||^2}{||z||^2} - 1 \right| \le \delta_W, \ \forall z \in \mathbb{R}^n \tag{5}$$

Notice that $\sigma(W) = \sup_{z \in \mathbb{R}^n, z \ne \mathbf{0}} \frac{||Wz||}{||z||}$ is the spectral norm of $W$, i.e., the largest singular value of $W$. As a result, $\sigma(W^T W - I) = \sup_{z \in \mathbb{R}^n, z \ne 0} |\frac{||Wz||^2}{||z||^2} - 1|$. In order to enforce orthogonality to $W$ from an RIP perspective, one may wish to minimize the RIP constant $\delta_W$ in the special case $k = n$, which according to the definition should be chosen as $\sup_{z \in \mathbb{R}^n, z \ne 0} |\frac{||Wz||^2}{||z||^2} - 1|$ as from (5). Therefore, we end up equivalently minimizing the spectral norm of $W^T W - I$:

$$(\text{SRIP}) \qquad \lambda \cdot \sigma(W^T W - I). \tag{6}$$

It is termed as the Spectral Restricted Isometry Property (SRIP) regularization.

*The above reveals an interesting hidden link*: regularizations with spectral norms were previously investigated in [36, 37], through analyzing small perturbation robustness and Lipschitz constant. The spectral norm re-arises from enforcing orthogonality when RIP condition is adopted. But compared to the spectral norm (SN) regularization [36] which minimizes $\sigma(W)$, SRIP is instead enforced on $W^T W - I$. Also compared to [37] requiring the spectral norm of $W$ to be exactly 1 (developed for GANs), SRIP requires *all singular values of $W$ to be close to 1*, which is essentially **stricter** because the resulting $W$ needs also be *well conditioned*.

We again refer to auto differentiation to compute the gradient of (6) for simplicity. However, even computing the objective value of (6) can invoke the computationally expensive EVD. To avoid that, we approximate the computation of spectral norm using the power iteration method. Starting with a randomly initialized $v \in \mathbb{R}^n$, we iteratively perform the following procedure a small number of times (2 times by default) :

$$u \leftarrow (W^T W - I)v, v \leftarrow (W^T W - I)u, \sigma(W^T W - I) \leftarrow \frac{||v||}{||u||}. \tag{7}$$

With such a rough approximation as proposed, SRIP reduces computational cost from $\mathcal{O}(n^3)$ to $\mathcal{O}(mn^2)$, and is practically much faster for implementation.

## 4  Experiments on Benchmarks

First of all, we will base our experiments on several popular state-of-the-art models: ResNet[6, 40] (including several different variants), Wide ResNet[21] and ResNext[20]. For fairness, all pre-processing, data augmentation and training/validation/testing splitting are strictly identical to the original training protocols in [21, 6, 40, 20]. All hyper-parameters and architectural details remain unchanged too, unless otherwise specified.

We structure the experiment section in the following way. In the first part of experiments, we design a set of intensive experiments on CIFAR 10 and CIFAR-100, which consist of 60,000 images of size 32×32 with a 5-1 training-testing split, divided into 10 and 100 classes respectively. We will train each of the three models with each of the proposed regularizers, and compare their performance with the original versions, in terms of both final accuracy and convergence. In the second part, we further conduct experiments on ImageNet and SVHN datasets. In both parts, we also compare our best performer SRIP with existing regularization methods with similar purposes.

**Scheme Change for Regularization Coefficients**  All the regularizers have an associated regularization coefficient denoted by $\lambda$, whose choice play an important role in the regularized training process. Correspondingly, we denote the regularization coefficient for the $\ell_2$ weight decay used by original models as $\lambda_2$. From experiments, we observe that fully replacing $\ell_2$ weight decay with orthogonal regularizers will accelerate and stabilize training at the beginning of training, but will

negatively affect the final accuracies achievable. We conjecture that while the orthogonal parameter structure is most beneficial at the initial stage, it might be overly strict when training comes to the final "fine tune" stage, when we should allow for more flexibility for parameters. In view of that, we did extensive ablation experiments and identify a switching scheme between two regularizations, at the beginning and late stages of training. Concretely, we gradually reduce $\lambda$ (initially 0.1-0.2) to $10^{-3}$, $10^{-4}$ and $10^{-6}$, after 20, 50 and 70 epochs, respectively, and finally set it to zero after 120 epochs. For $\lambda_2$, we start with $10^{-8}$; then for SO/DSO regularizers, we increase $\lambda_2$ to $10^{-4}/5 \times 10^{-4}$, after 20 epochs. For MC/SRIP regularizers, we find them insensitive to the choice of $\lambda_2$, potentially due to their stronger effects in enforcing $W^T W$ close to $I$; we thus stick to the initial $\lambda_2$ throughout training for them. Such an empirical "scheme change" design is found to work nicely with all models, benefiting both accuracy and efficiency. The above $\lambda/\lambda_2$ choices apply to all our experiments.

As pointed out by one anonymous reviewer, applying orthogonal regularization will change the optimization landscape, and its power seems to be a complex and dynamic story throughout training. In general, we find it to show a strong positive impact at the early stage of training (not just initialization), which concurs with previous observations. But such impact is observed to become increasingly negligible, and sometime (slightly) negative, when the training approaches the end. That trend seems to be the same for all our regularizers.

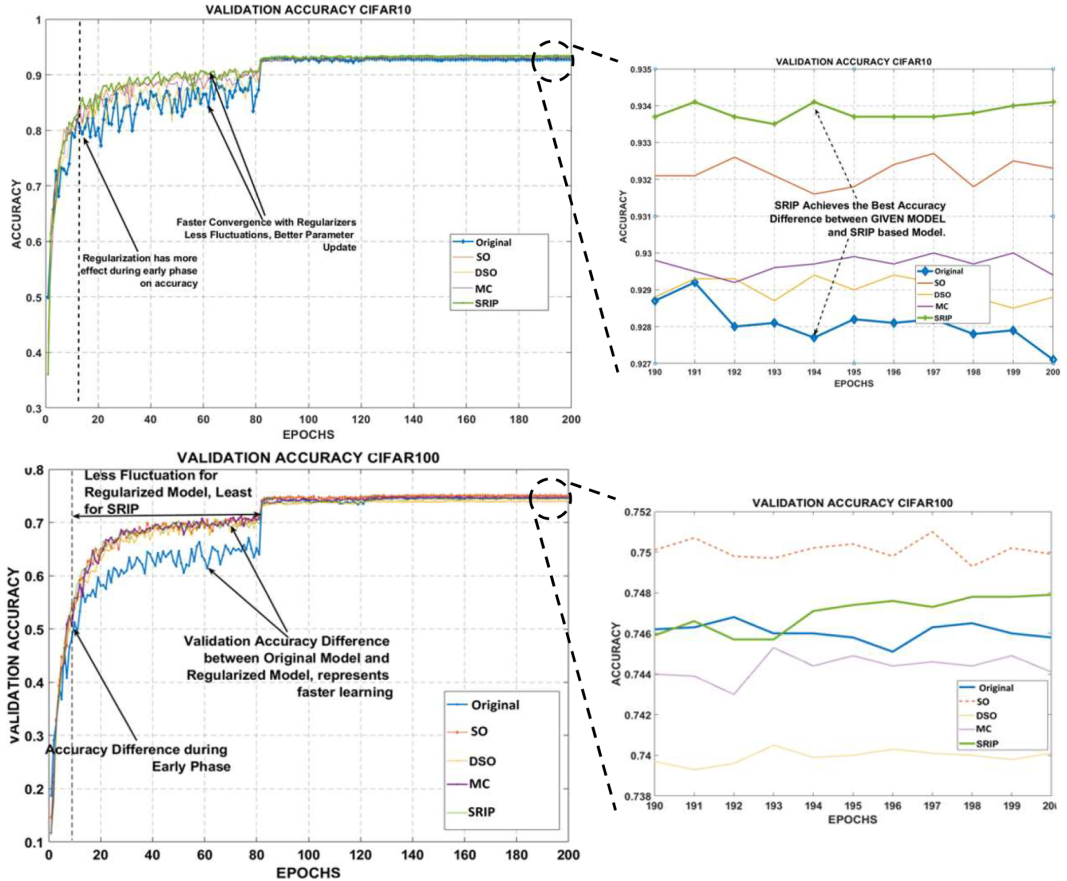

Figure 1: Validation curves during training for ResNet-110. Top: CIFAR-10; Bottom: CIFAR-100;

## 4.1 Experiments on CIFAR-10 and CIFAR-100

We employ three model configurations on the CIFAR-10 and CIFAR-100 datasets:

**ResNet 110 Model [6]** The 110-layer ResNet Model [6] is a very strong and popular ResNet version. It uses Bottleneck Residual Units, with a formula setting given by $p = 9n + 2$, where $n$ denotes the total number of convolutional blocks used and $p$ the total depth. We use the Adam optimizer to train the model for 200 epochs, with learning rate starting with 1e-2, and then subsequently decreasing to $10^{-3}$, $10^{-5}$ and $10^{-6}$, after 80, 120 and 160 epochs, respectively.

Table 1: Top-1 error rate comparison by ResNet 110, Wide ResNet 28-10 and ResNext 29-8-64 on CIFAR-10 and CIFAR-100. * indicates results by us running the provided original model.

| Model | Regularizer | CIFAR-10 | CIFAR-100 |
|---|---|---|---|
| ResNet-110 [6] | None | 7.04* | 25.42* |
| | SO | 6.78 | **25.01** |
| | DSO | 7.04 | 25.83 |
| | MC | 6.97 | 25.43 |
| | SRIP | **6.55** | 25.14 |
| Wide ResNet 28-10 [21] | None | 4.16* | 20.50* |
| | SO | 3.76 | 18.56 |
| | DSO | 3.86 | 18.21 |
| | MC | 3.68 | 18.90 |
| | SRIP | **3.60** | **18.19** |
| ResNext 29-8-64 [20] | None | 3.70* | 18.53* |
| | SO | 3.58 | 17.59 |
| | DSO | 3.85 | 19.78 |
| | MC | 3.65 | 17.62 |
| | SRIP | **3.48** | **16.99** |

**Wide ResNet 28-10 Model [21]** For the Wide ResNet model [21], we use depth 28 and $k$ (width) 10 here, as this configuration gives the best accuracies for both CIFAR-10 and CIFAR-100, and is (relatively) computationally efficient. The model uses a Basic Block B(3,3), as defined in ResNet [6]. We use the SGD optimizer with a Nesterov Momentum of 0.9 to train the model for 200 epochs. The learning rate starts at 0.1, and is then decreased by a factor of 5, after 60, 120 and 160 epochs, respectively. We have followed all other settings of [21] identically.

**ResNext 29-8-64 Model [20]** For ResNext Model [20], we consider the 29-layer architecture with a cardinality of 8 and widening factor as 4, which reported the best state-of-the-art CIFAR-10/CIFAR-100 results compared to other contemporary models with similar amounts of trainable parameters. We use the SGD optimizer with a Nesterov Momentum of 0.9 to train the model for 300 epochs. The learning starts from 0.1, and decays by a factor of 10 after 150 and 225 epochs, respectively.

**Results** Table 1 compares the top-1 error rates in the three groups of experiments. To summarize, SRIP is obviously the winner in almost all cases (except the second best for ResNet-110, CIFAR-100), with remarkable performance gains, such as an impressive **2.31%** top-1 error reduction for Wide ResNet-28-10. SO acts a surprisingly strong baseline and is often only next to SRIP. MC can usually outperform the original baseline but remains inferior to SRIP and SO. DSO seems the most ineffective among all four, and might perform even worse than the original baseline. We also carefully inspect the training curves (in term of validation accuracies w.r.t epoch numbers) of different methods on CIFAR-10 and CIFAR-100, with ResNet-110 curves shown in Fig. 1 for example. All starting from random scratch, we observe that all four regularizers significantly accelerate the training process in the initial training stage, and maintain at higher accuracies throughout (most part of) training, compared to the un-regularized original version. The regularizers can also stabilize the training in terms of less fluctuations of the training curves. We defer a more detailed analysis to Section 4.3.

Besides, we validate the helpfulness of scheme change. For example, we train Wide ResNet 28-10 with SRIP, but without scheme change (all else remains the same). We witness a $0.33\%$ top-1 error increase on CIFAR-10, and $0.90\%$ on CIFAR-100, although still outperforming the original un-regularized models. Other regularizers perform even worse without scheme change.

**Comparison with Spectral Regularization** We compare SRIP with the spectral regularization (**SR**) developed in [36]: $\frac{\lambda_s}{2}\sigma(W)^2$, with the authors' default $\lambda_s = 0.1$. All other settings in [36] have been followed identically. We apply the SR regularization to training the Wide ResNet-28-10 Model and the ResNext 29-8-64 Model. For the former, we obtain a top-1 error rate of **3.93%** on CIFAR-10, and **19.08%** on CIFAR-100. For the latter, the top-1 error rate is **3.54%** for CIFAR-10, and **17.27%** for CIFAR-100. Both are inferior to SRIP results from the same settings of Table 1.

**Comparison with Optimization over Multiple Dependent Stiefel Manifolds OMDSM** We also compare SRIP with OMDSM developed in [15], which makes a fair comparison with ours, on soft regularization forms versus hard constraint forms of enforcing orthogonality. This work trained Wide ResNet 28-10 on CIFAR-10 and CIFAR-100 and got error rates **3.73%** and **18.76%** respectively, both being inferior to SRIP (3.60% for CIFAR-10 and 18.19% for CIFAR-100).

**Comparison with Jacobian Norm Regularization**   A recent work [41] propounds the idea of using the norm of the CNN Jacobian as a training regularizer. The paper used a variant of Wide ResNet [21] with 22 layers of width 5, whose original top-1 error rate was **6.66%** on on CIFAR-10, and and reported a reduced error rate of **5.68%** with their proposed regularizer. We trained this same model using SRIP over the same augmented full training set, achieving **4.28%** top-1 error, that shows a large gain over the Jacobian norm-based regularizer.

## 4.2   Experiments on ImageNet and SVHN

We extend the experiments to two larger and more complicated datasets: ImageNet and SVHN (Street View House Numbers). Since SRIP clearly performs the best in the above experiments, among the proposed four, we will focus on comparing SRIP only.

**Experiments on ImageNet**   We train ResNet 34, Pre-ResNet 34 and ResNet 50 [40] on the ImageNet dataset with and without SRIP regularizer, respectively. The training hyperparameters settings are consistent with the original models. The initial learning rate is set to 0.1, and decreases at epoch 30, 60, 90 and 120 by a factor of 10. The top-5 error rates are then reported on the ILSVRC-2012 val set, with single

Table 2: Top-5 error rate comparison on ImageNet.

| Model | Regularizer | ImageNet |
|---|---|---|
| ResNet 34 [6] | None | 9.84 |
| | OMDSM [15] | 9.68 |
| | SRIP | **8.32** |
| Pre-Resnet 34 [40] | None | 9.79 |
| | OMDSM [15] | 9.45 |
| | SRIP | **8.79** |
| ResNet 50 [6] | None | 7.02 |
| | SRIP | **6.87** |

model and single-crop. [15] also reported their top-5 error rates with both ResNet 34 and Pre-ResNet 34 on ImageNet. As seen in Table 2. SRIP clearly outperforms the best for all three models.

**Experiments on SVHN**   On the SVHN dataset, we train the original Wide ResNet 16-8 model, following its original implementation in [21] with initial learning 0.01 which decays at epoch 60,120 and 160 all by a factor of 5. We then train the SRIP-regularized version with no change made other than adding the regularizer. While the original Wide ResNet 16-8 gives rise to an error rate of **1.63%**, SRIP reduces it to **1.56%**.

## 4.3   Summary, Remarks and Insights

From our extensive experiments with state-of-the-art models on popular benchmarks, we can conclude the following points:

- In response to the question in our title: *Yes, we can gain a lot from simply adding orthogonality regularizations into training*. The gains can be found in both final achievable accuracy and empirical convergence.

  For the former, the three models have obtained (at most) 0.49%, 0.56%, and 0.22% top-1 accuracy gains on CIFAR-10, and 0.41%, 2.31%, and 1.54% on CIFAR-100, respectively. For the latter, positive impacts are widely observed in our training and validation curves (Figure 1 as a representative example), in particular faster and smoother curves at the initial stage. Note that those impressive improvements are obtained with no other changes made, and is extended to datasets such as ImageNet and SVHN.

- With its nice theoretical grounds, SRIP is also the best practical option among all four proposed regularizations. It consistently performs the best in achieving the highest accuracy as well as accelerating/stabilizing training curves. It also outperforms other recent methods utilizing spectral norm [36], hard orthogonality [15], and Jacobian norm [41]

- Despite its simplicity (and potential estimation bias), SO is a surprisingly robust baseline and frequently ranks second among all four. We conjecture that SO benefits from its smooth form and continuous gradient, which facilitates the gradient-based optimization, while both SRIP and MC have to deal with non-smooth problems.

- DSO does not seem to be helpful. It often performs worse than SO, and sometimes even worse than the un-regularized original model. We interpret it by recalling how the matrix $W$ is constructed (Section 3 beginning): enforcing $W^T W$ close to $I$ has "inter-channel" effects (i.e., requiring different output channels to have orthogonal filter groups); whereas enforcing $WW^T$ close to $I$ enforce "intra-channel" orthogonality (i.e., same spatial locations across

different filter groups have to be orthogonal). The former is a better accepted idea. Our results on DSO seems to provide further evidence (from the counter side) that orthogonality should be primarily considered for "inter-channel", i.e., between columns of $W$.

- MC brings in certain improvements, but not as significantly as SRIP. We notice that (4) will approximate (3) well only when $W$ has unit columns. While we find minimizing (4) generally has the empirical results of approximately normalizing $W$ columns, it is not exactly enforced all the time. As we observed from experiments, large deviations of column-wise norms could occur at some point of training and potentially bring in negative impacts. We plan to look for re-parameterization of $W$ to ensure unit norms throughout training, e.g., through integrating MC with weight normalization [24], in future work.

- In contrast to many SVD-based hard orthogonality approaches, our proposed regularizers are light to use and incur negligible extra training complexity. Our experiments show that the per-iteration (batch) running time remains almost unchanged with or without our regularizers. Additionally, the improvements by regularization prove to be stable and reproducible. For example, we tried to train Wide ResNet 28-10 with SRIP from three different random initializations (all other protocols unchanged), and find that the final accuracies very stable (deviation smaller than $0.03\%$), with best accuracy $3.60\%$.

## 5   Conclusion

We presented an efficient mechanism for regularizing different flavors of orthogonality, on several state-of-art convolutional deep CNNs [21, 6, 20]. We showed that in all cases, we can achieve better accuracy, more stable training curve and smoother convergence. In almost all times, the novel SRIP regularizer outperforms all else consistently and remarkably. Those regularizations demonstrate outstanding generality and easiness to use, suggesting that orthogonality regularizations should be considered as standard tools for training deeper CNNs. As future work, we are interested to extend the evaluation of SRIP to training RNNs and GANs. Summarizing results, a befitting quote would be: *Enforce orthogonality in training your CNN and by no means will you regret!*

**Acknowledgments**

The work by N. Bansal, X. Chen and Z. Wang is supported in part by NSF RI-1755701. We would also like to thank all anonymous reviewers for their tremendously useful comments to help improve our work.

## Footnotes

[1] https://github.com/nbansal90/Can-we-Gain-More-from-Orthogonality

[2]We also tried to first normalize columns of $W$ and then apply (4), without finding any performance benefits.

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
