[Reviews · NeurIPS 2018]

Reviewer 1



Update after rebuttal: The rebuttal has shown that the experiments were done to a rigorous, high standard, and further, reports new experiments with several larger scale data sets. If these results hold more generally, the findings could be widely used. I recommend acceptance. ______________________ Summary: This paper explores methods for regularizing deep convolutional networks towards orthogonal weight matrices. In extensive experiments with state of the art models, the paper shows that soft orthogonality can improve training stability and yield better classification accuracy than the same models trained without such regularization. The paper proposes a method to approximately enforce all singular values of the weight matrices to be equal to 1, using a sampling-based approach that does not require computing an expensive SVD operation. Major comments: This paper presents interesting experiments showing that regularization towards orthogonal weights can stabilize and speed up learning, particularly near the beginning of training; and improve final test accuracy in several large models. These results could be of broad interest. One concern with the experimental methods is that they use carefully sculpted hyper parameter trajectories for some methods. How were these trajectories selected? It is crucial that they be tuned on a validation set, not the final test set accuracy. The SRIP version of the orthogonality regularization is a clever way of seeking orthogonality through a truncated power iteration approach. This makes the regularization readily compatible with standard automatic differentiation, and could be widely used. The paper could be improved by adding wall clock time comparisons. How expensive are each of these regularizers? This could also highlight the benefits of the proposed approach relative to hard orthogonality constraints. The clarity of the paper is middling. The equations contain typos (eg, Eqn 5 is missing one side of the absolute value), and could be improved by a substantial editing effort. It would be useful to more explicitly describe the difference between initializing to scaled orthogonal weights vs enforcing orthogonality throughout training. The former is a preconditioning scheme that can cause later learning to go quickly _without changing the optimization landscape_. The latter is a different optimization (minimize the loss subject to orthogonal weights), which will change the optimization landscape. The results presented here seem to show a complex story regarding which approach is better. The ‘scheme change’ results suggest that orthogonal regularization is better used as a preconditioner, and harms accuracy if enforced at the end of learning. However this may not be true for SRIP. A careful examination of these results would be helpful.

Reviewer 2



SUMMARY This work investigates various possible soft orthogonality constraints, promoting feature diversity in convolution layers, and yielding some improvement in image classification on CIFAR-10 and CIFAR-100. STRENGTHS - Tested on popular classification models. - Pretty good overview of prior work. - SRIP is framed in an interesting way in equation 6, for use as a regularizer. - SRIP may be fairly cheap, computationally, when roughly approximated as proposed. CRITICISMS - Because the proposed constraints enforce filters in convolution layers to be orthogonal to each other, they do not enforce the convolutional operators to be orthogonal; rather, they promote feature diversity in convolution layers. This is indeed proposed in the referenced "all you need is a good init" (10) paper as a regularizer. Such a constraint does not target the condition number of layer or address vanishing or exploding signals through a layer. This must be clearly stated and discussed. - Regarding 'scheme change': is it useful to even have a weight decay term for non-convolutional layers? Weight decay in DNNs is in practice enforced as an L2 norm on the weights which is essentially a penalty on the spectral radius (it bounds the largest singular value). Results without scheme change are not shown. The utility of 'scheme change' is stated but not proven. - In section 3.1: if W is m*n, then W'W is n*n, not m*m. Furthermore, it is stated that m ≤ n is undercomplete and necessary for the |W'W-I| constraint to work but this is inaccurate. This is actually correct elsewhere in the text but in this section, some symbols appear incorrectly arranged. If m < n, then W may be overcomplete, restricting W'W is problematic, and WW' might work. If m > n, then it may be undercomplete and restricting W'W works. Finally, if m = n, then W may be complete (and restricting W'W works). -- "if and only if W is undercomplete (m ≤ n)" -> '(n ≤ m)' -- "For overcomplete W (m > n), its gram matrix ∈ m × m cannot be even close to identity, because its rank is at most n" -> '(n > m)' and 'rank is at most m' - Final performace values tend to be very close together. What is the variance when rerunning experiments? Rerunning an exact reproduction of wide resnet, variance contains the results of multiple papers that claimed SOTA over wide resnets. -- The authors state that MC outperforms the baseline with a 6.97% error vs 7.04% error on CIFAR 10 and 25.42% for both on CIFAR-100. The performance appears identical for both. - There is no description of the way validation data was split from training data. Was there always a validation set separate from the test set? This should be detailed and made clear. If testing was done on the test set (as in wide resnet), the evaluation is incorrect. - It would be useful to compare against a norm constraint on neighbouring activations ("Regularizing RNNs by Stabilizing Activations" - 2015) or a norm constraint on the gradients between neighbouring layers ("On the difficulty of training recurrent neural networks." - 2013). MINOR COMMENTS - Orthogonality constraints may be more useful for RNNs and GAN discriminators than for ResNets, which already have a stable gradient guarantee, or for encouraging feature diversity in convolution lahyers. - "which provides a potential link between regularizing orthogonality and spectrum." -- it's the same thing - "we gradually reduce λ (initially 0.1) by factors of 1e-3, 1e-4 and 1e-6" - not 'by factors of', rather 'to' - learning rate "decreased by a factor of 0.2" - 'by a factor of' means divided by, which would in this case mean the lr is increasing - similarly, 'by a factor of 0.1' - It's surprising that "DSO always outperforms selective regularization". - SRIP is interesting. There should be a discussion on the time cost of different regularization approaches. With such a rough approximation as proposed (2 power method iterations), SRIP reduces computational cost from O(n**3) for soft orthogonality to O(n**2); from matrix-matrix multiplication to matrix-vector multiplication. - Perhaps there is a better lambda schedule for soft orthogonality that would make it outperform SRIP. - The MC trick may need a higher weight at first than soft orthogonality and SRIP due to a sparser gradient signal. - If (4) is an approximation, it's not "equivalent". TYPOS - in Assumption 1: z in [R]^n - "but unnecessarily appropriate" - not necessarily - "phase(direction)" - space - "without bothering any assistance components." - strange wording - "could possibly to mutually" - 'to' -> 'be' - "structure is most benefit at the initial stage" - 'beneficial' - eq 5 missing vertical bar OVERALL OPINION The evaluation of which orthogonality regularizers are possible and useful, and by how much, is a useful one for the community. The SRIP formulation is interesting. However, the scope of the analysis should be expanded and the robustness of comparisons should be improved with statistical analysis. The effect of the proposed regularizations on learned representations and on network dynamics should be clearly identified and discussed. Some errors should be corrected. I would encourage submitting this work to a workshop. UPDATE Significant work was done in the rebuttal to address the criticisms. Although the authors did not address the significance of results in Table 1, they (a) confirmed the reliability of results in Table 2 and (b) performed additional experiments on both SVHN and ImageNet that go a long way toward convincing me that SRIP could be a reliably useful regularizer. The authors also included additional experiments with constraints on gradient norm change, further demonstrating superior results with SRIP. The authors demonstrated a will to correct errors and clarify misleading parts of the text. I would especially encourage that they follow through with clarifying in the beginning that the proposed work enforces orthogonality across filters, encouraging filter diversity, and does not enforce orthogonality "on linear transformations between hidden layers" (second paragraph) in the way that some of the references do. The authors clarified some concerns such as whether hyperparameters were tuned on the validation data. Authors responded qualitatively about time complexity -- this should be in big O notation and would work in their favor to advertise in the text, along with the qualitative note about perceived wall clock time, as the low cost of the method is attractive. In light of the additional results and the clarification going into the revision, I will raise my rating from a 4 to a 7.

Reviewer 3



This paper studied orthogonality regularizations on training deep CNNs. The authors introduced three novel regularization forms for orthogonality, using double-sided Frobenius norm, Mutual Coherence (MC) and Restricted Isometry Property (RIP) tools, respectfully. Those orthogonality regularizations can plug-and-play with training many CNNs without hassle. They are evaluated on three state-of-the-art models on CIFAR10/100 Strength: - The paper is easy to follow in general. MC and RIP are exploited from compressive sensing literature, and both of their combinations with CNN regularization seem new to me. Particularly, the authors provide an explanation of SRIP’s relationship with the recently proposed spectral norm regularization/spectral normalization [33,34], and shows SRIP to be stricter. - The proposed techniques are light-weight and easy to plug in. The extra complexities incurred seem to be small (using simplified methods auto-differentiation, power iteration, etc.). They could potentially be easily adopted by many models. - Experiment advantages are mostly consistent, sometimes notable (2.47%in CIFAR-100). SRIP also outperformed latest competitive methods [15,33]; I like Section 4.5 comparison in particular, which seems to concur that SRIP is stricter. Weakness: - The experiments are insufficient to validate the claim. Only CIFAR10/100 are used, but many of studied techniques that were effective on CIFAR10/100 and MNIST turned out ineffective on other larger datasets/tasks. I would be happy to raise my score if the authors could provide ImageNet improvement (at least for SO and SRIP). - As the authors also implied, MC is not enforced in the “right way” (columns not normalized). I would like the authors to report their MC performance with column normalization for completeness.